# Impedance Characteristics of Monolayer and Bilayer Graphene Films with Biofilm Formation and Growth

**DOI:** 10.3390/s22093548

**Published:** 2022-05-06

**Authors:** Ryoichi Nakagawa, Kai Saito, Hideyuki Kanematsu, Hidekazu Miura, Masatou Ishihara, Dana M. Barry, Takeshi Kogo, Akiko Ogawa, Nobumitsu Hirai, Takeshi Hagio, Ryoichi Ichino, Masahito Ban, Michiko Yoshitake, Stefan Zimmermann

**Affiliations:** 1National Institute of Technology (KOSEN), Suzuka College, Mie 510-0294, Japan; nakagawa.ryouichi@b.mbox.nagoya-u.ac.jp (R.N.); saito.k.ce@m.titech.ac.jp (K.S.); kougo@mse.suzuka-ct.ac.jp (T.K.); ogawa@chem.suzuka-ct.ac.jp (A.O.); hirai@chem.suzuka-ct.ac.jp (N.H.); 2Faculty of Medical Engineering, Suzuka University of Medical Science, Mie 510-0293, Japan; miura-h@suzuka-u.ac.jp; 3Nanomaterials Research Institute, National Institute of Advanced Industrial Science and Technology (AIST), Ibaraki 305-8565, Japan; m.ishihara@aist.go.jp; 4Department of Electrical & Computer Engineering, Clarkson University, Potsdam, NY 13699, USA; dmbarry@clarkson.edu; 5Science/Math Tutoring Center, State University of New York at Canton, Canton, NY 13617, USA; 6Institutes of Innovation for Future Society, Nagoya University, Nagoya 464-8601, Japan; hagio@mirai.nagoya-u.ac.jp (T.H.); ichino.ryoichi@material.nagoya-u.ac.jp (R.I.); 7Department of Applied Chemistry, Nippon Institute of Technology, Saitama 345-8501, Japan; ban@nit.ac.jp; 8National Institute for Materials Science (NIMS), Tsukuba 305-0044, Japan; yoshitake.michiko@nims.go.jp; 9Institute of Electrical Engineering and Measurement Technology, Leibniz Universität Hannover, 30167 Hannover, Germany; zimmermann@geml.uni-hannover.de

**Keywords:** biofilm, sensors, impedance, EPS, Raman spectroscopy

## Abstract

Biofilms are the result of bacterial activity. When the number of bacteria (attached to materials’ surfaces) reaches a certain threshold value, then the bacteria simultaneously excrete organic polymers (EPS: extracellular polymeric substances). These sticky polymers encase and protect the bacteria. They are called biofilms and contain about 80% water. Other components of biofilm include polymeric carbon compounds such as polysaccharides and bacteria. It is well-known that biofilms cause various medical and hygiene problems. Therefore, it is important to have a sensor that can detect biofilms to solve such problems. Graphene is a single-atom-thick sheet in which carbon atoms are connected in a hexagonal shape like a honeycomb. Carbon compounds generally bond easily to graphene. Therefore, it is highly possible that graphene could serve as a sensor to monitor biofilm formation and growth. In our previous study, monolayer graphene was prepared on a glass substrate by the chemical vapor deposition (CVD) method. Its biofilm forming ability was compared with that of graphite. As a result, the CVD graphene film had the higher sensitivity for biofilm formation. However, the monolayer graphene has a mechanical disadvantage when used as a biofilm sensor. Therefore, for this new research project, we prepared bilayer graphene with high mechanical strength by using the CVD process on copper substrates. For these specimens, we measured the capacitance component of the specimens’ impedance. In addition, we have included a discussion about the possibility of applying them as future sensors for monitoring biofilm formation and growth.

## 1. Introduction

Biofilms are inhomogeneous film-like matters formed by bacterial activities. Bacteria in various environments attach to materials’ surfaces and form biofilms on them, which could result in various industrial, hygiene, and medical problems. These problems cover broad areas including corrosion, scale problems (in pipe works, heat exchangers, toilets, kitchens, and bathrooms), and medical fronts in hospitals [1,2,3,4,5,6,7]. Therefore, it is very important to solve problems induced by biofilms. As for the evaluation of biofilms, many methods have already been proposed. They are mainly classified into two types. One of them is visualization by using expensive cutting-edge apparatus such as electron microscopes [8,9,10,11,12], laser microscopes [13,14,15,16,17], Raman or infrared analytical apparatus and a variety of their combinations [18,19,20,21,22,23,24,25,26,27,28,29]. These evaluation methods could provide valuable quantitative information to help us improve our evaluation of biofilms. However, the qualitative analyses have been difficult to use for obtaining quantitative information. Moreover, from a practical point of view, the method would be too professional/sophisticated for everyone to use. The other category includes biological methods such as measuring the number of bacteria, staining, gene analyses, etc., [7,30,31,32,33,34,35]. Among them, measuring the number of bacteria is not always compatible with biofilm formation and growth. The reason for this is that biofilms are formed by bacterial activities in the initial stage and their behaviors are affected by EPS during the growth stage. Gene analyses require special apparatus and professional knowledge. On the other hand, staining methods are visible, convenient, and do not require lots of professional skills to obtain a quantitative evaluation. However, they are not suitable for in situ monitoring. Based on this background information, it is desirable to sense (detect) biofilms by using electronic devices to overcome the weak points of conventional evaluation methods. 

In this experiment, we focused on the impedance behaviors of materials forming biofilms. When biofilms form on materials, the impedance behaviors change due to the different dielectric characteristics of the substrates. However, the susceptibility of materials (electrodes) to biofilm formation and growth is critical. In the past, we confirmed the relatively high susceptibility to biofilm formation [31,36,37,38,39]. In those cases, we used monolayer graphene on copper substrates produced by CVD [40], since the process could produce reproducible and high quality (high covering ratio) graphene films. However, the mechanical strength of monolayer graphene is relatively weak and the coverage is generally not so high. Therefore, in this experiment, we used double-layer graphene and carried out an investigation to find out if its impedance characteristics (mainly the capacitance components) would provide measurements suitable for practical sensing in the future.

## 2. Materials and Methods

### 2.1. Specimens—Graphene Film Formation

Copper foils of 0.21 × 0.247 m were heated to 600 °C by resistance heating. Then plasma formed in the gas mixture (of methane and hydrogen) and was irradiated on the specimens to form monolayer and bilayer graphene on copper substrates. The display of surface layers is shown in Figure 1, schematically. The graphene layer forms on 10 μm copper foil. The thickness of graphene films were confirmed by their transparency, since the transparency for monolayer and bilayer graphene were already fixed [40,41]. On the other hand, the 100 μm polyester film (detachable by heating) exists on the other side of the copper foil. Those sheets were cut into small coupon of 10 mm × 10 mm and were used for various tests.

### 2.2. Biofilm Formation and Freeze Dehydration 

To produce biofilms on specimens, we used Gram-negative *Escherichia coli* (*E. coli*, K-12, G6). The bacteria were cultivated in LB solution at 37 °C for 24 h in advance, so that the number of bacteria would reach 10^8^ CFU/mL (CFU stands for colony formation unit and corresponding to the colony numbers). Then graphene film specimens were immersed into each of 12 plastic wells and 1.6 mL bacterial solution (precultured as mentioned above) were added into each well. The plastic wells were put into an incubator at 25 °C for 24 h. 

After 24 h, the specimens immersed in LB culture were freeze dried in the following way. The mixed solutions of distilled water and ethanol were prepared at various ratios (water: ethanol = 7:3, 5:5, 3:7, 2:8, 1:9, 0.5:9.5, 0.2:9.8, and 0:1) to replace water components of biofilms completely with ethanol. Specimens were immersed in each solution for 15 min in the order of increasing ethanol ratios. Then the specimens were immersed into each of mixed solutions of ethanol and t-butyl alcohol in the order of 7:3, 5:5, 3:7, 2:8, 1:9, 0.5:9.5, 0.2:9.8, 0:1 of ethanol and t-butyl solutions to completely replace ethanol in biofilms with t-butyl solution. Finally, the specimens were kept frozen in a freezer for 30 min and then dried in a vacuum desiccator by using a rotary pump. Then the specimens were analyzed using Raman spectroscopy.

### 2.3. Raman Spectroscopy

The apparatus used for Raman spectroscopy was NRS-3100 made by JASCO Corporation, Tokyo, Japan. The optical microscope was combined with the spectroscopy. A green laser beam (100 mW, 532 nm) was irradiated on the specimens’ surfaces and Raman shifts were observed between 500 and 4000 cm**^−^**^1^. The measurements were repeated several times at certain locations for each of three specimens (N = 3).

### 2.4. Absorbance Measurement

After the impedance measurements, specimens were immediately stained by 0.1% crystal violet (CV) solution. The 1.6 mL CV solution was prepared for each plastic well and specimens were immersed into them for 30 min. Then the CV solution was removed, and the specimens were washed three times with pure water. Next, they were immersed into 3 mL ethanol solution for 30 min at room temperature to extract CV-stained biofilms. Then 200 μL of extracted solutions underwent absorbance measurements by a plate reader (Multi Scan FC, Thermofisher Scientific Co, Yokohama, Japan.). Light (570 nm**^−^**^1^) was irradiated to each extracted solution and the absorbance was measured to evaluate the quantitative amount of biofilm.

### 2.5. Impedance Measurement

As for the impedance measurement, specimens were removed from the plastic wells after being immersed in bacterial solutions. They dried naturally for 15 min in the air and then had impedance measurements. The electrode part and the schematic circuit are shown in Figure 2. Specimens were put between the electrode parts of the measurement system and sandwiched between two copper plate electrodes. The contact part between the copper electrodes and specimens was composed of vinylidene chloride films (10 micrometer thickness) and the sandwiched electrodes were secured by clips, so that a certain pressure was softly applied to make the electric contact. The electrode part was connected to a chemical impedance Analyzer (IM 3590, Hioki Co., Ueda, Japan). The impedance characteristics were measured when the frequencies from 100 Hz to 20 kHz were applied to the specimens.

## 3. Results

### 3.1. Raman Spectroscopy with Optical Microscopy

Generally, bilayer graphene films covered the copper foil substrate better and the sensitivity of graphene to biofilm formation was expected to be higher. The mechanical strength for two-layer graphene films was also expected to be higher than that for monolayer graphene films.

Figure 3 shows the Raman spectroscopic results for monolayer graphene film specimen and two-layer graphene film specimen. For one of the representative monolayer graphene specimens (Figure 3a), the remarkable peak was found at 2700 cm^−1^ and it could be attributed to 2D band peak of graphene. A G band peak was observed at 1580 cm^−1^. It shows the existence of graphite. However, the peak was weak. On the other hand, one of the representative bilayer graphene specimens (Figure 3b) showed the G-band peak at 1580 cm^−1^ more clearly and suggests that the crystallographic characteristics of graphene film would have been lost from the viewpoint of crystallography to some extent. 

Figure 4 shows the Raman spectroscopic peaks for specimens forming biofilms. Figure 4a shows a typical results for monolayer graphene film specimens, while Figure 4b shows the results for a typical bilayer graphene film specimen. For specimens forming biofilms on monolayer graphene, Raman peaks derived from biofilms were found. For example, the peak(s) around 2500 cm*^−^*^1^ could be attributed to lipids and/or proteins. From 1000 cm*^−^*^1^ to 1600 cm*^−^*^1^, the peaks could be attributed to proteins, lipids, and/or nucleic acids. On the other hand, the 2D band peak around 2700 cm*^−^*^1^ became relatively weak and almost disappeared. Moreover, for specimens forming biofilms on bilayer graphene films, some peaks attributed to biofilms appeared in addition to the 2D and G-band Raman peaks, as shown in Figure 4b. For example, lipid and/or protein peaks were found at 2800 cm*^−^*^1^. Furthermore, peaks from 1400 cm*^−^*^1^ to 1500 cm*^−^*^1^ (peaks assigned to proteins and/or lipids) were found. They clearly show that biofilms were qualitatively formed on graphene film specimens.

### 3.2. Biofilm Assay Using CV Staining

It was hard and impossible for us to quantitatively evaluate biofilm amounts by using Raman Spectroscopy. Then we measured and evaluated stained biofilms quantitatively, using the CV staining method. Figure 5 shows the results of CV staining. The vertical axis shows the absorbance, which corresponds to the amount of biofilm formed on the specimens. As shown in Figure 5, the amount of biofilm (absorbance values) was higher for monolayer graphene film specimens (Figure 5a) than for bilayer graphene film specimens (Figure 5b), even though the graphene coverage of the monolayer graphene film specimen was lower than that of the bilayer film.

### 3.3. Impedance Measurements

Figure 6 shows the impedance behaviors for specimens of graphene films without biofilms. Figure 6a shows the change of impedance behavior with increasing frequencies. Figure 6b shows the differential curve of impedance. 

As shown in Figure 6, the impedance value decreased with increased frequencies due to dielectric relaxation. The differential curve shows that the tendency and the differential value increased with frequencies monotonously. However, the increment gradually became smaller with increased frequency. In the range of lower frequencies, the differential curve was vibrated. We presume, it could be attributed to superfluous noises sometimes seen in the region. 

Figure 7 shows the impedance curve with frequencies (Figure 7a) for the monolayer specimen on which biofilms formed. The impedance values increased in the range of higher frequencies. The differential curve (Figure 7b) shows the complicated peaks at higher frequencies. We presume the differential peaks could be attributed to the existence of various polymeric substances derived from biofilms. 

Figure 8a shows the impedance behavior with frequencies for the bilayer graphene film specimen without biofilms. The tendencies for the impedance curve and the differential curve (Figure 8b) were almost the same, even though the absolute values were different from each other. 

Figure 9a shows the impedance curve for a specimen of bilayer graphene film and Figure 9b shows the differential curve. The bilayer graphene film shows the same tendencies for the impedance curve and the differential curve with those for specimens of monolayer graphene film. Therefore, the various peaks observed in the range of high frequencies should be attributed to EPS in the same way. 

Figure 10 shows the impedance behaviors for freeze-dried specimens of bilayer graphene film. Figure 10a shows the impedance curve, while Figure 10b shows the differential curve. In this case, the impedance and the differential curves were very similar to those of specimens without biofilms. 

## 4. Discussion

We found in our previous studies that the monolayer graphene would be sensitive to biofilm formation, when compared with graphite specimens. It could be attributed to the existence of π electrons perpendicular to the surface in the resonance state. Therefore, biofilms formed on graphene films rather easily. This was also true for bilayer graphene films. The existence of biofilms was shown concretely by Raman Spectroscopy. As described in the previous section, the impedance measurement shows that the increase of values and peaks of differential curves were found in higher frequencies. We presume that those increases, and peaks could be attributed to the biofilm formation and growth. Particularly, we expect that the differential peaks would be utilized as biofilm sensors for the future.

The comparison between non-freeze dried and freeze-dried specimens suggests that the peaks in the higher frequency area could be related closely to the adsorption of hydroxyl groups to EPS of biofilms. The information was lost for the freeze-dried specimens. 

Figure 11 shows the schematic diagram for biofilm on specimens and the electric path change with frequencies. At lower frequencies, the current tends to flow in the lower concentration areas of EPS. Then the ionic conduction in the aqueous solution of biofilms would occur more easily. However, the current tends to move in higher concentration areas of EPS when the frequency increases. Therefore, the impedance must change in higher frequencies and the differential peaks could be found there. Therefore, we could conclude that the differential peaks at higher frequencies are essential to indicate the existence of biofilms and could be used as biofilm sensors in the future.

As for the impedance characteristics of electrodes, Figure 12 shows the corresponding equivalent circuits for biofilms. We presume at this point that the conductive solution in biofilms could be attributed to the part composed of only a resistance. The part composed of only a capacitance could correspond to the electrode part. Then the central part composed of parallel capacitance and resistance corresponds to the biofilm itself. The resistance component represents the ionic conductive aqueous part, and the capacitance is the insulated part composed of EPS. Therefore, the multiple peaks found in the higher frequency area would correspond to various EPSs. Examples include polysaccharides, proteins, lipids, and nucleic acids, respectively. This presumption should be confirmed by further quantitative and qualitative investigations. 

## 5. Conclusions

By using monolayer and bilayer graphene films on copper foils and applying impedance spectroscopy to the specimens, we investigated the possibility of sensing (detecting) for biofilm formation and growth. The graphene film specimens were sandwiched by insulating electrodes and the impedance behaviors. We observed their changes with frequencies. The following results were obtained.

(1)Specimens’ impedance decreased with the increase of frequency before biofilm formation. (2)As for graphene film specimens with biofilms, the increase of impedance was obtained in the relatively higher frequency area. (3)The phenomenon could be confirmed for specimens of both monolayer and bilayer graphene films. (4)The increase of impedance values and the differential peaks found in relatively higher frequencies were not observed for the freeze-dried specimens. It suggests that the peaks would be related closely to the adsorption of hydroxyl groups to the EPS of biofilms.(5)The differential curves of impedance behaviors show some peaks in the higher frequency area. The peaks would correspond to the formation of EPS components. Therefore, the peaks could be utilized for the sensing of biofilms.

## Figures and Tables

**Figure 1 sensors-22-03548-f001:**
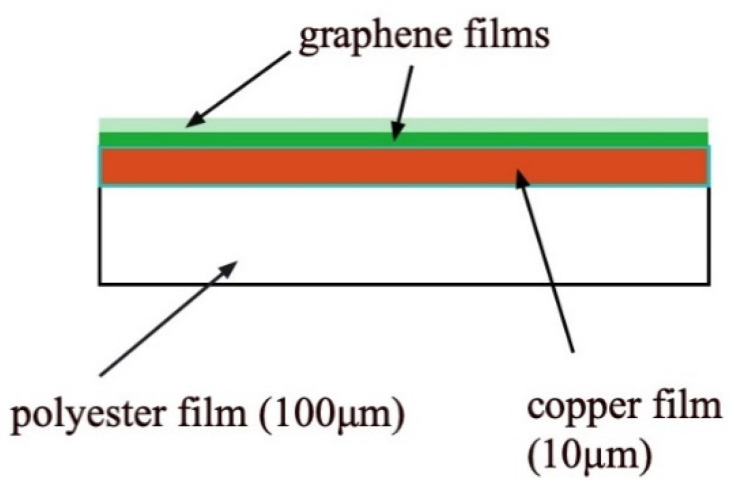
The structure of the specimen used in this experiment.

**Figure 2 sensors-22-03548-f002:**
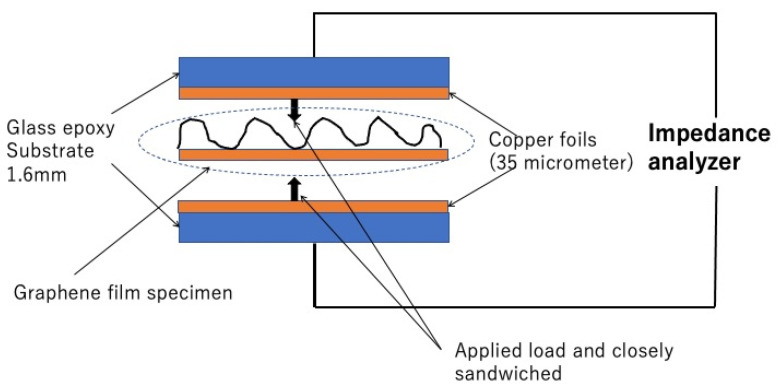
The electrode structure for impedance measurements.

**Figure 3 sensors-22-03548-f003:**
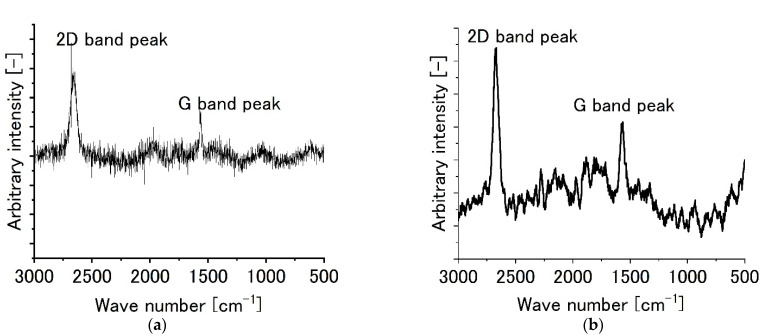
Raman peaks for graphene film specimens without biofilms. (**a**) Monolayer graphene film, (**b**) bilayer graphene film.

**Figure 4 sensors-22-03548-f004:**
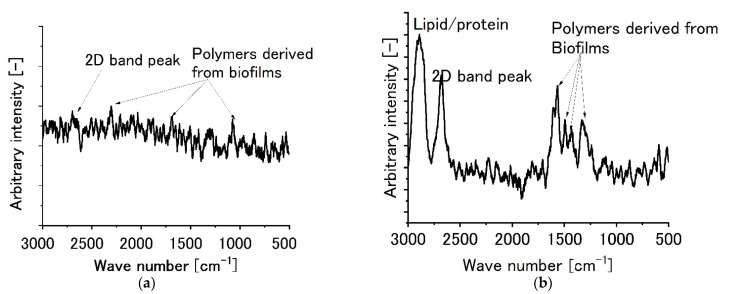
Raman peaks for graphene film specimens with biofilms. (**a**) Monolayer graphene film, (**b**) two-layer graphene film.

**Figure 5 sensors-22-03548-f005:**
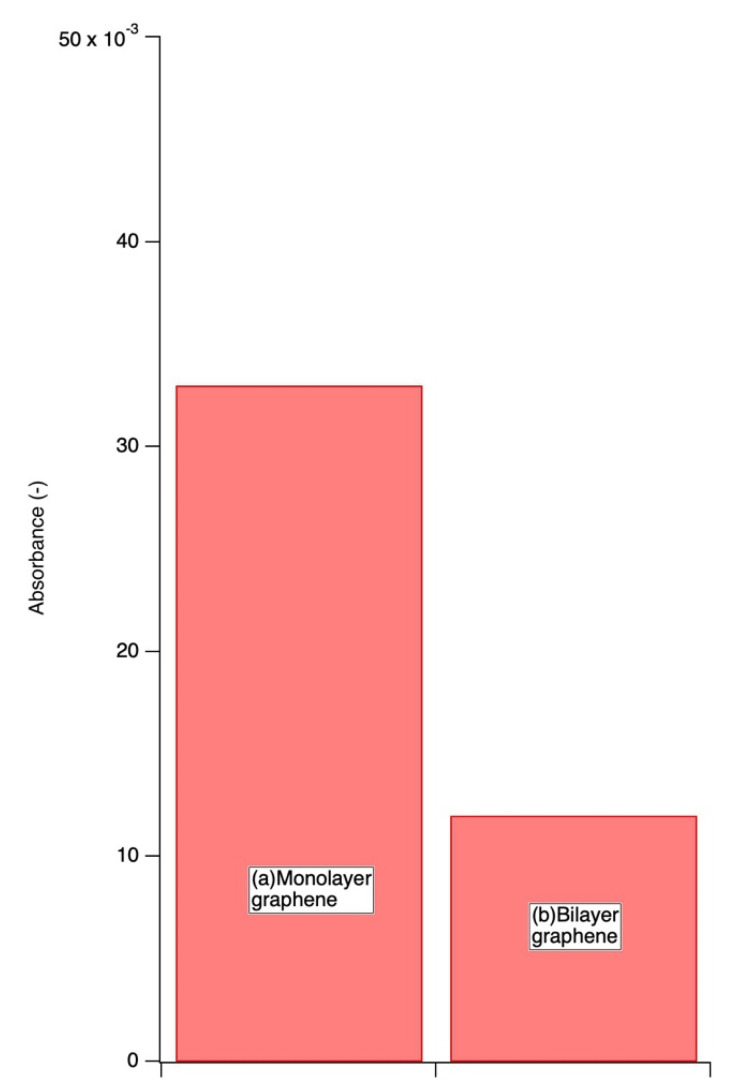
Absorbances of specimens stained by 0.1% crystal violet solution.

**Figure 6 sensors-22-03548-f006:**
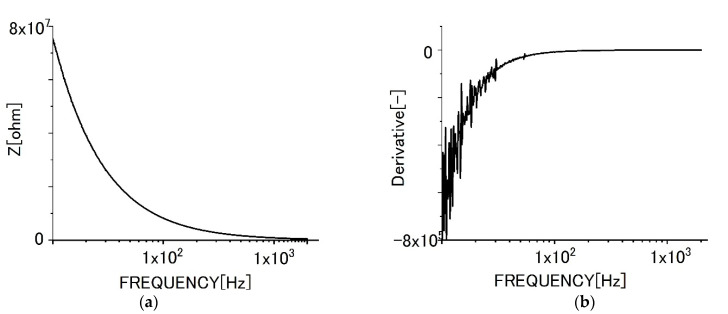
Impedance behavior for specimens of monolayer graphene film (**a**) impedance curve, (**b**) differential curve.

**Figure 7 sensors-22-03548-f007:**
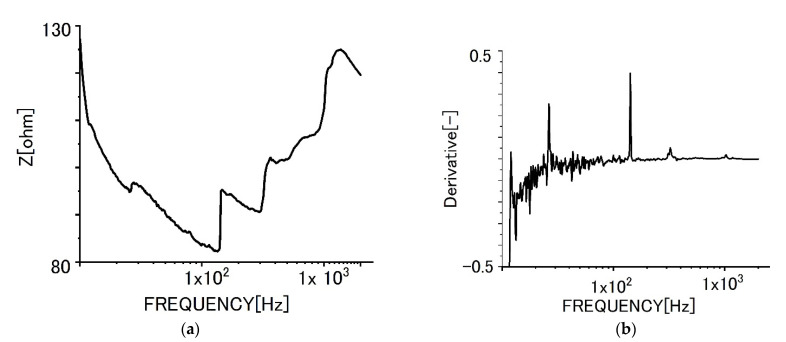
Impedance behaviors of monolayer graphene film specimens with biofilms. (**a**) Impedance curve, (**b**) differential curve.

**Figure 8 sensors-22-03548-f008:**
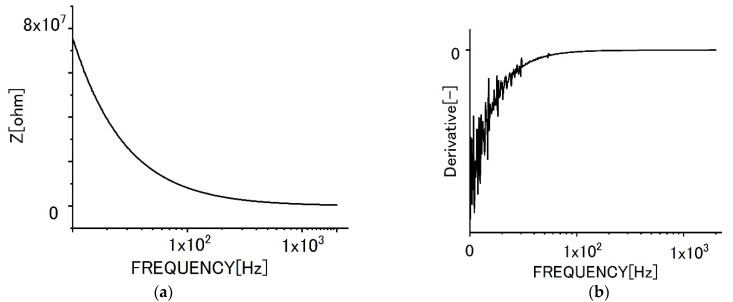
Impedance behaviors for specimens of bilayer graphene film. (**a**) Impedance curve, (**b**) differential curve.

**Figure 9 sensors-22-03548-f009:**
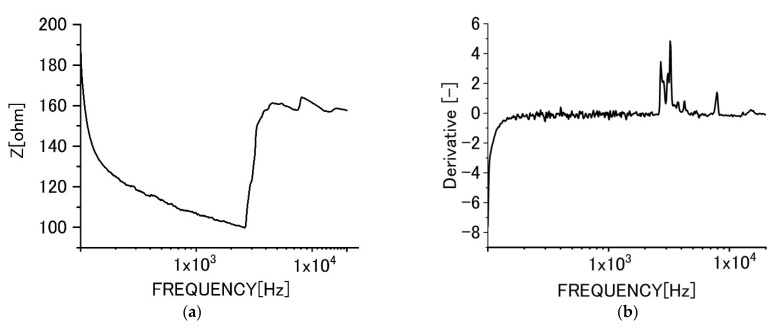
impedance behaviors of bilayers graphene film specimens with biofilms, (**a**) impedance curve, (**b**) differential curve.

**Figure 10 sensors-22-03548-f010:**
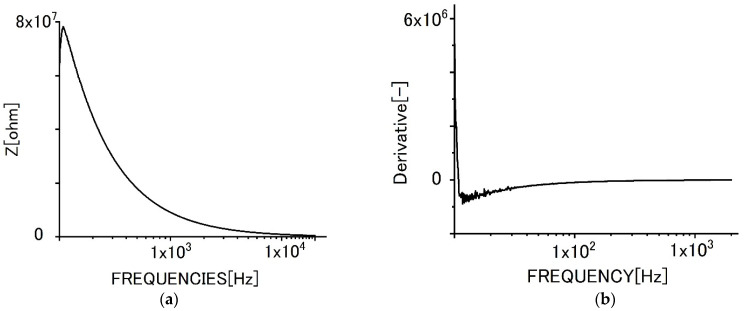
Impedance behaviors of freeze-dried bilayer graphene film specimens with biofilms. (**a**) Impedance curve, (**b**) differential curve.

**Figure 11 sensors-22-03548-f011:**
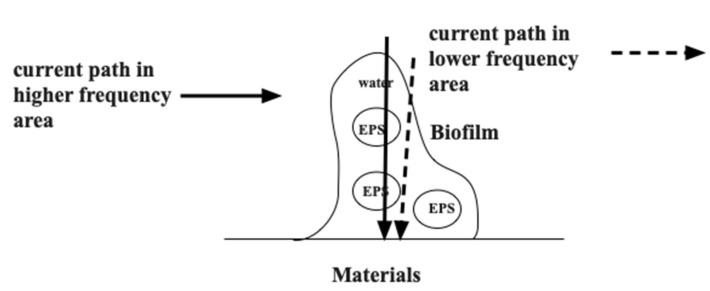
Schematic diagram of biofilms and current paths.

**Figure 12 sensors-22-03548-f012:**
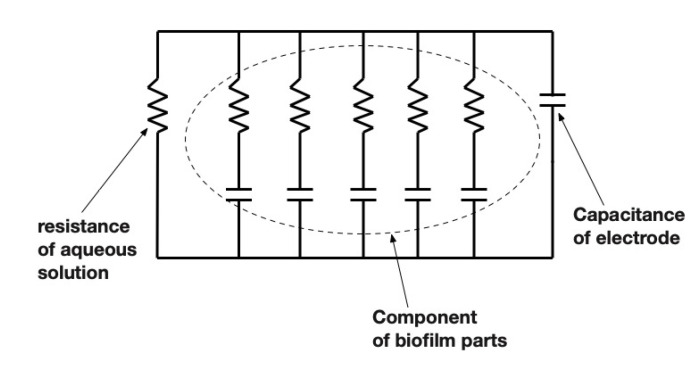
Equivalent circuit of this model.

## Data Availability

Not applicable.

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
