# Peer review of "Impedance Characteristics of Monolayer and Bilayer Graphene Films with Biofilm Formation and Growth"

_sensors, 2022, doi:10.3390/s22093548_

Round 1
Reviewer 1 Report
Journal Name: Sensors Title: Impedance Characteristics of single-layer and two-layer Graphene Films with Biofilm Formation and Growth The authors have made detailed research on “single and bilayer graphene biofilm impedance characterstics”. The article fits the scope of the journal but the article can be furtherimproved. I recommend its publication with major revision. Here I am mentioning my detailed comments. 1. Instead of two layer author can use bilayer term. 2. Why authors used CVD to deposit graphene 3. How authors measured thickness of the graphene layers 4. Authors should be careful while writing all sentences should be properly allighned, take care of the font style and size, few paragraphs are bolded without any reasons. 5. The quality of the Fig.6-10 can be improved 6. What are the error limits of the experiments that have to be explained? 7. Please check the grammatical and syntax errorAuthor Response
Dear editors and reviewers:
Hello, I am Hideyuki Kanematsu, the corresponding author for this manuscript entitle “Impedance Characteristics of single-layer and bilayer Graphene Films with Biofilm Formation and Growth”.
Thank you so much for your valuable and precious revision ideas, tips and critics. We accepted all of them into our revised manuscript and we believe that the revised one would be suitable for the publication. Let us mention and describe how we revised the manuscript. The corresponding places are marked in yellow color in the manuscript directly, as well as the descriptions mentioned below.
ⅠFor reviewer1.
1.The term “two-layer” was changed to “bilayer” in the title as well as figures and text. At the same time, we changed “single-layer” to “monolayer”, so that both would be expressed in compatible ways.
- the reason for why we used CVD.
We used CVD just as an example and we believe it could be applied to any methods for graphene production. However, we needed the reproducible and painstaking production process for this experiment. One of our corresponding authors, Dr. Ishihara and his colleagues have developed a novel CVD process in AIST, Japan. Therefore, we used it for this experiment. To clarify this, we cited one of their scientific papers and added the explanation about the reason in the text. (Line 89-90).
- How did authors measure the thickness.
A certain value of transparency corresponds to one layer of graphene. We cited one of the papers showing the result and explained about it in the text. Concretely speaking, we estimated the thickness using transparency in the following way. When one layer graphene forms, the transparency decreases 97.7%. When the two layers of graphene form, the decrease of transparency is equal to 97.7 x 97.7 %....and so on.
- Font style, size etc.
We revised all of them. Sorry and thank you for your kind tips.
- We tried to improve the quality of figures (Fig.6-10) as much as possible. As for Fig.4-(2) was wrong in the previous version, we replaced it with the correct one.
- In this case, we confirmed the tendency, for example, the increase of impedance at higher frequencies etc. was absolutely the same and repeated. However, the absolute value itself was not fixed. We measured the impedance behavior qualitatively in this case. Quantitative analysis should be the next step experiment, adjusting the experimental conditions quite the same completely. Therefore, the error, dispersion and other statistic factors are out of focus.
- All sentences were checked and revied thoroughly. However, all sentences in the text were checked in advance by one of our English authors (Prof. Dana M. Barry.). She has been a capable English scientific editor in Clarkson University and got the award for the editorial works in 10 consecutive years. She checked and revised all of our manuscripts in our first submission version. Then she assured us that English level was originally acceptable. However, for the sake of precaution, she checked all of revised version once again and revised many sentences. They were marked in yellow color.
Reviewer 2 Report
Dear Authors,
the work requires a thorough reconstruction. There are many errors in the manuscript. The work is not prepared according to the journal's requirements. Please reorganize the manuscript according to the journal's requirements (there are errors in writing the literature, conclusions, different fonts, ...). It is unacceptable to me in this form.
Author Response
Dear editors and reviewers:
Hello, I am Hideyuki Kanematsu, the corresponding author for this manuscript entitle “Impedance Characteristics of single-layer and bilayer Graphene Films with Biofilm Formation and Growth”.
Thank you so much for your valuable and precious revision ideas, tips and critics. We accepted all of them into our revised manuscript and we believe that the revised one would be suitable for the publication. Let us mention and describe how we revised the manuscript. The corresponding places are marked in yellow color in the manuscript directly, as well as the descriptions mentioned below.
>The work requires thorough reconstruction. There are many errors….
We revised all sentences and incorporated your (reviewers’ comments) into our text. We believe the work was reconstructed as a result once again. Thank you for your useful suggestions. Literature style was united into MDPI’s format and revised thoroughly.
>The work requires thorough reconstruction. There are many errors….
We revised all sentences and incorporated your (reviewers’ comments) into our text. We believe the work was reconstructed as a result once again. Thank you for your useful suggestions. Literature style was united into MDPI’s format and revised thoroughly.
Reviewer 3 Report
Please see the attachment.

Author Response
Dear editors and reviewers:
Hello, I am Hideyuki Kanematsu, the corresponding author for this manuscript entitle “Impedance Characteristics of single-layer and bilayer Graphene Films with Biofilm Formation and Growth”.
Thank you so much for your valuable and precious revision ideas, tips and critics. We accepted all of them into our revised manuscript and we believe that the revised one would be suitable for the publication. Let us mention and describe how we revised the manuscript. The corresponding places are marked in yellow color in the manuscript directly, as well as the descriptions mentioned below.
- Fonts, styles of texts were thoroughly revised, as shown with yellow markers.
- Escherichia coli
We revised the term E.coli to Escherichia coli at line 106.
- We revised all “degrees Celsius” to ℃.
- The iteration of the same unit at line 136
We changed it to 500 and 4000 cm-1.
- CFU was explained at Line number 109-123. On the other hand, BF stands for biofilm(s)
But the description at line 131 was the only one case where BF was used. Therefore, “BF” at line 131 was changed to biofilms.
- “CFU/m” was changed to “CFU/mL” at line 109.
- The “ml” was corrected as “mL”.
- “Fig.” was changed to “Figure” in the whole sentences.
- The expressions of all references were corrected according to MDPI’s expression, as shown in the file.
- The subtitle of section 2.4 was corrected according to your tips. (line 140)]
- “Ionic liquid” at line 258 was changed to “solution”.
Round 2
Reviewer 1 Report
The authors made sufficient changes article can be accepted in the present form